# Antiproliferative Activity and DNA Interaction Studies of a Series of N4,N4-Dimethylated Thiosemicarbazone Derivatives

**DOI:** 10.3390/molecules28062778

**Published:** 2023-03-20

**Authors:** Serena Montalbano, Annamaria Buschini, Giorgio Pelosi, Franco Bisceglie

**Affiliations:** 1Department of Chemistry, Life Sciences and Environmental Sustainability, University of Parma, Parco Area Delle Scienze 11/A, 43124 Parma, Italy; 2COMT (Interdepartmental Centre for Molecular and Translational Oncology), University of Parma, Parco Area Delle Scienze 11/A, 43124 Parma, Italy; 3CERT (Center of Excellence for Toxicological Research), University of Parma, via Gramsci 14, 43126 Parma, Italy

**Keywords:** thiosemicarbazone metal complexes, U937 cell line, antiproliferative activity, DNA damage, DNA interaction

## Abstract

The exploitation of bioactive natural sources to obtain new anticancer agents with novel modes of action may represent an innovative and successful strategy in the field of medicinal chemistry. Many natural products and their chemical analogues have been proposed as starting molecules to synthesise compounds with increased biological potential. In this work, the design, synthesis, and characterisation of a new series of N4,N4-dimethylated thiosemicarbazone Cu(II), Ni(II), and Pt(II) complexes are reported and investigated for their in vitro toxicological profile against a leukaemia cell line (U937). The antiproliferative activity was studied by MTS assay to determine the GI_50_ value for each compound after 24 h of treatment, while the genotoxic potential was investigated to determine if the complexes could cause DNA damage. In addition, the interaction between the synthesised molecules and DNA was explored by means of spectroscopic techniques, showing that for Pt and Ni derivatives a single mode of action can be postulated, while the Cu analogue behaves differently.

## 1. Introduction

In the design and synthesis of next-generation anticancer agents, thiosemicarbazones (TSCs) represent a class of compounds with a remarkable pharmacological profile [1,2,3]. TSCs are obtained through a condensation reaction of a thiosemicarbazide with an aldehyde or a ketone [4] to give rise, due to their chemical structure, to a wide range of coordination modes [5,6]. Different TSC derivatives can be obtained by introducing substituents on the ligand backbone, and, in particular, by inserting substituents on the thioamide and hydrazine nitrogen atoms, a procedure that gives rise to a variety of interesting molecules [7]. Some structural features that appear to be essential for the biological activity of TSC have been identified: in the thiocarbonyl group, the exchange of the sulphur with selenium or oxygen; the change of the attachment point of the TSC moiety in the parent aldehyde or ketone; and the substitution on the terminal N4 position [8]. Other factors include electron density distributions, the nature of the substituents, the geometry and symmetry of the starting ligand, the metal binding ability, the solubility, and the possibility of interaction with the cell membrane [9].

Due to their versatility, TSCs display a broad spectrum of biological activities such as being antifungal [10,11], antibacterial [12], and antiviral [13]; the biological profiles could be enhanced by complexation with metal ions and the active compounds are normally planar molecules consisting of a pyridine ring in an N,N,S tridentate system [6]. Furthermore, a number of TSC derivatives were synthesised and reported as potential anticancer agents both in vitro [14] and in vivo [15], as free ligands [16], and as metal complexes [17], against several cancer types.

TSCs are considered versatile pharmacophores due to their ability to form stable complexes with transition metals, particularly with nickel [18,19,20,21]; copper [22,23,24,25]; platinum [26]; zinc [27]; and gold [28]. Metal ion coordination could affect the TSC’s ability to cross the plasmatic membrane; metal ions can, in fact, hide the polar part of TSC around themselves in such a way that the ligand molecule exposes the hydrophobic moiety outwards, thus facilitating a cellular internalisation [6]. As a result, the metal complexes tend to show more bioactivity compared with the corresponding free ligands [5], and they can also reduce the side effects of the organic parent compound [8].

Although TSCs action mechanisms have not been completely elucidated, it is well-known that most of them have multiple molecular and cellular targets and interfere with several pathways. Different mechanisms have been proposed to explain TSCs’ anticancer activity. For instance, TSCs may act by forming redox active complexes and producing reactive oxygen species (ROS) that affect the activity of enzymes such as topoisomerase I and II [29], tyrosinase [30], and metalloproteinase [31]. Furthermore, anticancer activities induced by TSCs affect several proteins, inducing apoptosis via the mitochondrial pathway of cancer cells and arresting the cell cycle [32]. The key role of TSCs as anticancer compounds is often related to their capability to inhibit the enzyme ribonucleotide reductase, interfering with the essential di-iron tyrosyl radical centre of the small enzymatic subunit. Currently, the most promising TSC as an anticancer agent is Triapine, a highly hydrophobic tridentate derivate with chelating properties [33], which shows great potentialities for the treatment of several types of cancer. It has been evaluated as a single agent in more than 30 clinical phase I/II trials including advanced leukaemia [34], head and neck squamous cell carcinoma [35], and renal cell carcinoma [36]. There are also several clinical trials to evaluate this drug in combination therapy with other anticancer agents, such as cisplatin [37], gemcitabine [38], cytarabine [39], doxorubicin [40], irinotecan [41], or radiation [37,42].

In our previous studies, citronellal and other natural compounds have been chosen as starting aldehydes to synthesise different TSCs and their corresponding metal complexes, with the aim of investigating how substituent groups could influence their biological activity. We have published several papers containing the syntheses and characterisations of three different TSC derivatives, resulting from citronellalthiosemicarbazonate (tcitr): nickel ([Ni(tcitr)_2_]) [18,19,20], copper ([Cu(tcitr)_2_]) [25], and platinum complexes ([Pt(tcitr)_2_]) [20]. Our previous experiments highlighted that all the compounds possessed interesting antiproliferative activity and caused a significant DNA damage, inducing structural alterations and demonstrating a direct interaction with chromatin in human cancer cells. In the present work, we describe the synthesis, the chemical characterisation, and an evaluation of the antiproliferative activity of novel N4,N4-dimethyl citronellal thiosemicarbazone derivatives as anticancer agents. We also performed in vitro studies to detect genotoxic potential and to identify potential interactions of these metal complexes with DNA.

## 2. Results

### 2.1. Chemistry

The newly synthesised compounds were obtained in good yields. The metal complexes exhibited a square planar geometry (Figure 1), in agreement with analogous complexes obtained with similar ligands [20].

### 2.2. Cytotoxicity of Newly Synthesised Molecules on U937 Cells

In our previous works, we investigated the antiproliferative activity of [Ni(tcitr)_2_], [Pt(tcitr)_2_], and [Cu(tcitr)_2_] after 24 h treatment against the leukaemia cell line U937, identifying a GI_50_ (50% Growth Inhibition) value equal to 10.0 µM [22], 7.0 µM, and 33.0 µM [20], respectively (Table 1). Cytotoxic profiles of [Ni(tcitr)_2_] obtained against a panel of different cancer cells have already been reported in a previous research [22].

Starting from citronellal TSC derivates, we synthesised and characterised three different dimethylated complexes ([Ni(4dm-(tcitr)_2_], [Pt(4dm-(tcitr)_2_], and [Cu(4dm-(tcitr)_2_]). In this study, we investigated their antiproliferative activity on leukemic cell line U937, a model already used in our studies. Cells were treated for 24 h with six different concentrations of the compounds, from 0.5 to 100.0 µM, to determine dose–response curves. The growth inhibiting concentrations at 50% (GI_50_) were established for each compound and the data were expressed as the mean of three different experiments, as shown in Table 1 (Table 1). The dose–response curve in U937 cells revealed that all the dimethylated complexes showed a mild antiproliferative activity after 24 h treatment (Figure 2): GI_50_ values were, broadly speaking, numerically similar across all compounds, with values around 30 µM (Table 1). The antiproliferative profile of these compounds seems to be independent of the metal ion used in the coordination process. Statistically significant differences among treatments are reported in Table 2.

### 2.3. Genotoxicity of Newly Synthesised Molecules on U937 Cells

In a previous work, we investigated the DNA damage induced by the nonmethylated [Ni(tcitr)_2_], [Pt(tcitr)_2_], and [Cu(tcitr)_2_] on U937 cells after 1 h treatment. In this context, in order to find out whether the introduction of a methyl group could alter the genotoxic potential of the starting molecules, the Alkaline Comet Assay was performed on U937 cells treated with the dimethylated complexes.

A preliminary check of cell number and viability was performed by staining U937 cells with Trypan blue. We did not observe a significant decrease in viability after 1 h treatment with the dimethylated complexes; therefore, all the treated cells were processed through Alkaline Comet Assay.

All the newly synthesised compounds already induced DNA damage in leukocytes after 1 h treatment: [Ni(4dm-(tcitr)_2_] > [Pt(4dm-(tcitr)_2_] > [Cu(4dm-(tcitr)_2_]. The percentage of DNA in the comet tail (Tail Intensity %) has been considered a useful parameter to identify the genotoxic potential of the metal complexes.

In particular, [Ni(4dm-(tcitr)_2_] was the most genotoxic compound, causing a significant DNA migration in the Comet assay at low concentration, starting from 10 µM (Figure 3). [Pt(4dm-(tcitr)_2_] produced DNA damage in U937 cells at 20 and 50 µM. At the highest experimental concentration (50 µM), the percentage of the tail intensity induced by [Pt(4dm-(tcitr)_2_] was almost double compared with that of [Ni(4dm-(tcitr)_2_]. [Cu(4dm-(tcitr)_2_] caused a strong and significant increase in DNA migration only at 50 µM (Figure 3).

In Figure 4, we report a comparison between the DNA damage induced by the nonmethylated and the dimethylated metal complexes. All the compounds showed a genotoxic potential, but the starting molecules were more active (Figure 4).

### 2.4. DNA Interaction Studies

If simple groove binding and electrostatic interactions with small molecules show little perturbations or no perturbation at all on the CD spectra on the base stacking and helicity bands [43], this cannot be said for the three complexes under study. Nevertheless, it is difficult to identify the proper mode of action towards DNA. It is noteworthy that for [Ni(4dm-(tcitr)_2_] and [Pt(4dm-(tcitr)_2_], we observed the presence of an isodichroic point that allowed us to think that the metal complex interacted in a single way towards the nucleic acid, as already noted in the nonmethylated complexes [25]. Comparing the CD spectra of the three compounds, it can be said that as the compound concentration increases, the effect is deeper. As far as the Pt complex is concerned, it has been shown that the intensity of the positive CD band yielded by B-DNA at ca 275 nm is increased as a consequence of DNA modification to relatively low r values ([complex]/[DNA] < 0.05) by the complexes containing the cis-[PtCl_2_(amine)_2_] unit. On the other hand, the modification of DNA by cisplatin at higher rb (>0.1) decreases this positive band. It has been suggested that the enhancement of the CD band at 275 nm due to the modification of DNA by the complexes containing cis-[PtCl_2_(amine)_2_] unit reflects distortions of a non-denaturational nature (hydrogen-bond breakage) [43]. The reduction of this CD band induced by the binding of platinum complexes is consistent with the occurrence of short single-stranded segments containing unpaired bases (denatured regions) [44]. Therefore, the same behaviour can be postulated for our Pt complex (Figure 5), even if at r values higher than those observed for cis-[PtCl_2_(amine)_2_] units. The behaviour observed for [Ni(4dm-(tcitr)_2_] was slightly different (Figure 6).

The CD spectrum of DNA exhibits a monotonous decrease in both band intensities, more evident for the one at lower nm. This effect can be associated with an induced change of DNA conformation that is not well understood. As far as [Cu(4dm-(tcitr)_2_] is concerned, the absence of an isodichroic point suggests that the metal complex can interact with DNA in more than one way. For this complex, the major effect is on the positive band, therefore implied in base stacking, while the negative band, implied in helicity, has a strong variation too, but only starting from r = 0.5 (Figure 7).

To further unravel the mechanisms behind the interactions between DNA and the compounds under study, thermal denaturation profiles of DNA were studied. The melting temperature (Tm) of DNA incubated with the compounds was compared with that of native CT-DNA Tm (86 °C). At the studied concentration, only complex [Ni(4dm-(tcitr)_2_] showed a slight effect on the melting temperature (+0.7 °C), suggesting its capability to induce a slight stabilisation of the DNA duplex. For all the other compounds, no significant increase in Tm was observed. Therefore, from those experiments, intercalation can be substantially excluded.

## 3. Discussion

In this work, we report a study concerning the behaviour of coordination compounds of Cu, Ni, and Pt obtained with the methylated forms of previously studied compounds, and, in particular, we focused on the ability of TSC to interfere with DNA.

The syntheses of the newly produced ligand and complexes were afforded compounds in good yields. In analogy with similar compounds already obtained [20,22,25], we expected the compounds to have a square planar environment around the metal ion, which was bound to two deprotonated ligands via the sulphur atom and the iminic nitrogen. The desired stoichiometry was confirmed by means of mass spectrometry and elemental analysis, while the IR spectra, corroborated by the NMR data in the case of Ni and Pt only, were in accordance with the deprotonation of the ligands and the use of S and N as chelating atoms. As already observed, the substituents on the terminal nitrogen do not affect the coordinative behaviour of citronellalthiosemicarbazones, and therefore the different biological activities observed are imputable to the role of the different metal ions.

We then investigated the antiproliferative profile of these novel TSC derivatives against a leukaemia cell line U937, the in vitro cell model system systematically used in our studies, to validate the toxicological profile of new potential anticancer agents [18,19,20,22]. The newly synthesised compounds displayed a mild antiproliferative effect after 24 h treatment: [Pt(4dm-(tcitr)_2_] > [Cu(4dm-(tcitr)_2_] > [Ni(4dm-(tcitr)_2_]. Overall, the tested compounds induced an inhibition of cell proliferation in a dose-dependent manner. The resulting GI_50_ values were in the micromolar range, indicating that the chemical scaffold of our molecules could be investigated as the starting point for the design of anticancer agents.

Our previous results on the antiproliferative activity of other metal complexes derived from citronellal TSC showed that after 24 h the most promising complexes were [Pt(tcitr)_2_] and [Ni(tcitr)_2_] (the sequence was [Pt(tcitr)_2_] > [Ni(tcitr)_2_] > [Cu(tcitr)_2_]) [18,20,22]. Comparing the GI_50_ values obtained from the dose–response curves, the introduction of the methyl groups in the skeleton of the starting molecules seems to be relatable to a decrease in their antiproliferative effects (Table 1).

The factors that can account for these observed behaviours may be identified in two main contributions: uptake and interactions with DNA. We had envisaged that a decrease in polarity due to the methylations of the terminal amino groups would have had an impact on the polarity of the molecules, by increasing their lipophilicity and, as a consequence, we expected an increase in the uptake and also an increase in the activity. Unexpectedly, we observed that the GI_50_s of all three complexes increased and flattened to about the same order of magnitude, while in the unmethylated compounds, the Pt and Ni derivatives were by far more active. Curiously enough, the GI_50_ of the Cu complex remained almost the same. From this viewpoint, against our expectations, a decreased polarity did not improved the activity of our compounds. The other interesting point is that the methylation of the terminal amino group also quenched the activity of the Pt and Ni derivatives, bringing them to the same level as Cu. This brings to mind the fundamental role of the amino group in the stabilisation of the cisplatin adducts with DNA. It is known that the presence, in proximity of the metal centre, of an amino group able to form hydrogen bonds contributes to the stabilisation of the metal complex–DNA interactions, probably forming hydrogen bonds with the oxygens of the phosphate groups in the DNA backbone. The methylation could also have been responsible for the weaker interactions between the Ni and Pt complexes with DNA and for the flattening of the GI_50_ values.

A broad spectrum of DNA damage was detected with the Alkaline Comet Assay. Treated cells had a viability higher than 70%, as confirmed by Trypan blue staining. All the dimethylated complexes already caused DNA damage after 1 h treatment (Figure 3 and Figure 4). In particular, the nickel complex induced a significant increase in DNA breaks at low concentrations. We obtained a similar result in U937 cells treated with [Ni(tcitr)_2_] [18,19].

Although [Pt(4dm-(tcitr)_2_] is the most antiproliferative metal complex, we observed DNA damage only at the highest tested concentrations. If we compare the results obtained between the parental and the dimethylated compounds, the first one showed a greater genotoxic potential. The platinum complexes probably interacted with DNA, but they could act through several action mechanisms. Several studies reported that cisplatin-like metallodrugs can bind DNA bases and cause changes in DNA conformation, which subsequently block DNA replication and transcription, triggering apoptosis [45,46]. However, there are other mechanisms of interaction with DNA: platinum derivatives can produce intercalation by stacking between the DNA bases, noncovalent and/or electrostatic interactions, hydrophobic or hydrogen bonding, and even cleavage of the DNA helix.

The parental and the dimethylated copper complexes were genotoxic only at the highest concentration used in the assay. We presume that the DNA damage observed after the treatment with copper TSC derivatives could be due to an excessive production of ROS species, due to the fact that copper could present different oxidation states in the cell. Oxidative stress due to ROS is known to cause DNA lesions of both a single- and double-strand (DSB) nature through the direct interaction of ROS with DNA [47].

Our results indicate that the dimethylated compounds are less genotoxic than the parent molecules [19,20] (Figure 3 and Figure 4). The DNA damage certainly demonstrates that the molecules cross the cell membrane and reach the nucleus. DNA could therefore represent a possible target, but further studies are required to understand if the metal complexes could act as multitarget agents: their cyto- and genotoxic effects could be involved in several cellular pathways.

## 4. Materials and Methods

4,4-dimethyl-3-thiosemicarbazide, 98% (TCI), disodium salt of calf thymus DNA (CT DNA) (Serva), and Platinum(II) chloride were purchased from Sigma-Aldrich (St. Louis, MI, USA); nickel(II) acetate tetrahydrate and copper(II) chloride dihydrate were obtained from Carlo Erba; and (S)-citronellal was obtained from Alfa Aesar. All solvents for the syntheses were obtained from Sigma-Aldrich.

^1^H NMR were recorded on a Bruker Anova spectrometer at 300 MHz, with chemical shift reported in δ units (ppm) (Bruker, Billerica, MA, USA). NMR spectra were referenced relative to residual NMR solvent peaks. The FT-IR measurements were recorded on Perkin Elmer’s (Waltham, MA, USA) Spectrum Two in the 4000–400 cm^−1^ range, equipped with the ATR accessory. The shapes and signal intensities are reported as w (weak), m (medium), s (strong), sh (sharp), and b (broad). Elemental analyses were performed using Flashsmart CHNS Elemental Analyser (Thermofisher Scientific, (Waltham, MA, USA). Mass analyses were carried out using a Waters (Milford, MA, USA) Acquity Ultraperformance ESI-MS spectrometer with a Single Quadrupole Detector (Mode used: Flow Injection; Source temperature (°C) 150; Desolvation Temperature (°C) 300; Cone Gas Flow (L/Hr) 100; Desolvation Gas Flow (L/Hr) 480; Solvent Flow (mL/min) 0.2; Capillary voltage (kV) 3, Cone voltage (V) 20/50/80). The compounds were dissolved in MeOH. Melting points were determined using a SPM3 apparatus (Stuart Scientific South San Francisco, CA, USA). Circular dichroism spectra were recorded with a Jasco J-715 spectropolarimeter (Jasco, Tokyo, Japan). UV-Vis spectra were collected using Thermofisher Scientific’s Evolution 260 Bio Spectrophotometer in a quartz cuvette.

### 4.1. Synthesis and Chemical Characterisation

[Ni(tcitr)_2_] and [Cu(tcitr)_2_] were synthesised and characterised following a procedure reported previously by Belicchi Ferrari [48], while [Pt(tcitr)_2_] was synthesised and characterised as reported by Bisceglie [20].

#### 4.1.1. Synthesis of 4dm-Htcitr

A solution of S-citronellal (0.35 mL, 1.95 mmol) was added to a refluxing solution of 4,4-dimethylthiosemicarbazide (0.232 g, 1.95 mmol) in EtOH 95% (20 mL), with stirring. After 20 min under stirring at reflux temperature, the mixture was left to cool down to room temperature, and the round-bottomed flask containing the mixture was transferred to an ultrasonic bath and left at 40 °C for 60 min. The solution was then poured into a crystalliser and the solvent was allowed to evaporate at room temperature. An oily product was obtained that was redissolved in 95% ethanol and dried by rotavapor. A sticky yellow–orange solid was obtained which was characterised.

Yield: 70%.

FT IR: 3204 cm^−1^ ν NH; 2961, 2914, 2867, 2851 cm^−1^ ν CH aliph.; 1552 cm^−1^ ν C=N; 746 cm^−1^ ν C=S.

ESI-MS: 256 MH^+^, 255 M^+^, 294 MK^+^.

EA for C_13_H_25_N_3_S: calculated C (61.13%), H (9.87%), N (16.45%), S (12.55%); found C (61.42%), H (10.01%), N (16.55%), S (12.34%).

¹H NMR (400 MHz, DMSO D6) δ 5.07 (1H, m, Hb), 3.01 (6H, m, Hk), 2.55 (2H, m, Hg), 1.89 (2H, t, Hc), 1.78 (1H, m, He), 1.63 and 1.50 (6H, 2s, 3H each, Ha), 1.36 and 1.29 (2H, 2m, 1H each, Hd), 0.88 (3H, m, Hf) (Figure 8).

#### 4.1.2. Synthesis of [Ni(4dm-(tcitr)_2_]

In a round-bottomed 50 mL flask, 4dm-Htcitr (0.305 g, 1.20 mmol) were dissolved in 20 mL of 95% ethanol with the help of magnetic stirring and heating. A solution of nickel (II) acetate (0.149 g, 0.60 mmol) in 25 mL of 95% ethanol was separately prepared. The metal salt solution was then added drop by drop under stirring to the one of the ligand, and the resulting mixture was left under stirring for about 30 min. A change in colour of the solution from blue–green to brown was observed. The solution of the complex was poured in a crystalliser and left to evaporate at room temperature. An oily dark brown solid was obtained which was characterised.

FT IR: 2960, 2915, 2861 cm^−1^ ν CH aliph.; 1528 cm^−1^ ν C=N.

ESI-MS: 566 MH^+^, 703 MK^+^.

EA for C_26_H_48_N_6_NiS_2_: calculated C (55.02%), H (8.52%), N (14.81%), S (11.30%); found C (55.24%), H (8.78%), N (14.55%), S (11.10%).

¹H NMR (400 MHz, DMSO D6) δ 5.07 (1H, m, Hb), 3.03 (6H, m, Hk), 2.80 and 2.68 (2H, 2m, 1H each, Hg), 1.94 (2H, m, Hc), 1.77 (1H, m, He), 1.65 and 1.56 (6H, 2s, 3H each, Ha), 1.41 and 1.20 (2H, 2m, 1H each, Hd), 0.89 (3H, m, Hf).

#### 4.1.3. Synthesis of [Pt(4dm-(tcitr)_2_]

In a round-bottomed 50 mL flask, 4dm-Htcitr (0.380 g, 1.49 mmol) were dissolved in 30 mL of 95% ethanol with the help of magnetic stirring and heating. A solution of platinum (II) chloride (0.198 g, 0.75 mmol) in the minimum quantity of DMSO was separately prepared. The metal salt solution was then added drop by drop under stirring to the one of the ligand, and the resulting mixture was left under stirring for about 30 min. A change in colour of the solution from pale yellow to brown was observed. The solution of the complex was poured into a crystalliser and left to evaporate at room temperature. A solid powder was obtained which was characterised.

FT IR: 2958, 2918, 2867 cm^−1^ ν CH aliph.; 1535 cm^−1^ ν C=N; 738 cm^−1^ ν C=S.

ESI-MS: 704 MH^+^.

EA for C_26_H_48_N_6_PtS_2_: calculated C (44.36%), H (6.87%), N (11.94%), S (9.11%); found C (44.71%), H (6.64%), N (12.16%), S (9.08%).

¹H NMR (400 MHz, DMSO D6) δ 5.04 (1H, m, Hb), 3.10 (6H, m, Hk), 2.85 and 2.77 (2H, 2m, 1H each, Hg), 1.95 (2H, m, Hc), 1.77 (1H, m, He), 1.63 and 1.56 (6H, 2s, 3H each, Ha), 1.40 and 1.21 (2H, 2m, 1H each, Hd), 0.87 (3H, m, Hf).

#### 4.1.4. Synthesis of [Cu(4dm-(tcitr)_2_]

In a round-bottomed 50 mL flask, 4dm-Htcitr (0.380 g, 1.62 mmol) were dissolved in 30 mL of 95% ethanol with the help of magnetic stirring and heating. A solution of copper (II) chloride dihydrate (0.139 g, 0.81 mmol) in 20 mL of water was separately prepared. The metal salt solution was then added dropwise under stirring to the one of the ligand, and the resulting mixture was left under stirring for about 30 min. A change in colour of the solution from pale blue to dark green was observed. The solution of the complex was poured in a crystalliser and left to evaporate at room temperature. A solid powder was obtained which was characterised.

FT IR: 2964, 2916, 2859 cm^−1^ ν CH aliph.; 1536 cm^−1^ ν C=N.

ESI-MS: 572 MH^+^.

EA for C_26_H_48_CuN_6_S_2_: calculated C (54.56%), H (8.45%), N (14.68%), S (11.20%); found C (54.78%), H (8.34%), N (14.85%), S (10.96%).

### 4.2. Cell Lines and Culture Conditions

U937 cells (ATCC, CRL-1593.2) were obtained from the American Tissue Culture Collection (Rockville, MD, USA) and were cultured in Roswell Park Memorial Institute Medium (RPMI-1640) (Euroclone, Italy), supplemented with 10% (*v*/*v*) foetal bovine serum (Euroclone, Italy), 1% L-glutamine (2 mM) (Euroclone, Italy), and 1% penicillin (100 U/mL)/streptomycin (100 μg/mL) (Euroclone, Italy). Flasks were maintained at 37 °C and 5% CO_2_ in a humidified atmosphere. Culture medium was refreshed every two or three days during sub-culturing, and the determination of cell number and viability was performed with the Trypan blue exclusion method. Briefly, cells were resuspended in complete medium and Trypan blue solution 0.4% (Gibco™, Thermofisher Scientific, Waltham, MA, USA) was added (ratio 1:1). One hundred cells for each experimental condition were counted manually using a haemocytometer.

### 4.3. In Vitro Antiproliferative Activity of Metal Complexes on Human Cancer Cells by MTS Assay

The antiproliferative activity of newly synthesised metal complexes was evaluated in vitro on U937 lines. In the exponential growth phase, cells were seeded at 5 × 10^4^/mL into 96-well flat-bottom microplates. Cells were cultured in complete medium supplemented with 5% FBS, 1% L-glutamine, and 1% penicillin/streptomycin at 37 °C, 5% CO_2_, in the absence of phenol red, according to the manufacturer’s protocol. After 24 h from seeding, different concentrations of compounds in the range, from 0.5 to 100.0 µM in a sterile DMSO solution, were added to the wells and incubated for 24 h. Negative control was obtained treating cells with DMSO. The antiproliferative activity was evaluated by a colorimetric assay (CellTiter96^®^ Aqueous One Solution Cell Proliferation Assay, Promega Corporation, Madison, WI, USA): 20 µL of CellTiter96^®^ AQueous One Solution Cell Proliferation Assay was added directly to culture wells, incubated for 4 h, and the absorbance recorded at 485 nm with a 96-well plate reader (TECAN SpectraFluor Plus, Männedorf, Switzerland). Cytotoxicity experiments were carried out in quadruplicate. The GI_50_ value for each molecule, concentration causing 50% reduction in the cell number in comparison with control cells, was calculated as reported by the US National Cancer Institute (NCI) 60 anticancer drug screen guidelines [49].

### 4.4. In Vitro Genotoxic Activity of Metal Complexes on U937 Cells through Alkaline Comet Assay

U937 cells were seeded 24 h before treatment at a concentration of 1 × 10^5^ cell/mL in 1 mL wells. Cells were treated with increased concentrations (1.0–5.0–10.0–20.0–50.0 µM) of the compounds for 1 h. Positive and negative controls were represented by ethylmethanesulfonate (EMS) [2 mM] and DMSO [100 μM], respectively. After treatment period at 37 °C, the determination of cell numbers and viabilities was performed with the Trypan blue exclusion method (see above). Only the treatments that had a viability higher than 70% were processed in the assay. Cells were transferred onto degreased microscope slides previously dipped in 1% normal melting agarose (NMA) for the first layer. The agarose was allowed to set for 20 min at 4 °C before the addition of a final layer of low melting agarose (LMA). Cell lysis was carried out at 4 °C overnight by exposing the cells to a buffer containing 2.5 M NaCl, 10 mM Na_2_EDTA, 10 mM Tris–HCl, 1% Triton X-100 and 10% DMSO, and pH 10. To detect single and double DNA strand breaks and alkali-labile sites, the electrophoretic migration was performed in an alkaline buffer (1 mM Na_2_EDTA, 300 mM NaOH, 0 °C) at pH > 13 (DNA unwinding: 20 min; electrophoresis: 20 min, 0.78 Vcm^−1^ 300 mA). Slides were then washed with a neutralisation solution (0.4 M Tris–HCl, pH 7.5).

DNA was stained with 75 µL ethidium bromide (10 µg/mL) before examination at 400 × magnification under a Leica DMLS fluorescence microscope (excitation filter BP 515–560 nm, barrier filter LP 580 nm), using an automatic image analysis system (Comet Assay IV—Perceptive Instruments Ltd. UK). Data are expressed as percentage of DNA in the tail region of the comet (TI%, tail intensity percentage). For each sample, coded and evaluated blind, 100 cells were analysed. The cells showing completely fragmented chromatin (i.e., hedgehog cells) were assessed as a further indicator of cytotoxicity. These cells were not evaluated by image analysis, but were recorded separately.

### 4.5. CD Titrations

Circular dichroism (CD) spectra were recorded at 25 1C on a Jasco J-715 spectropolarimeter with buffer compensation; each spectrum is the average of three independent measurements. One centimetre path-length quartz cuvettes were used. Disodium salt of calf thymus DNA (CT DNA) (Sigma) was used as received and stored at 4 °C. Solutions of DNA in 10 mM of PBS (pH = 7.4), 137 mmol^−1^·L NaCl, and 2.7 mmol^−1^·L KCl gave a ratio of the UV absorbances at 260 to 280 nm, A260/A280, of 1.9, indicating that the DNA was sufficiently free of protein. The concentrations of the stock solutions of DNA, expressed in moles of nucleotide phosphate [NP], were determined by UV absorbance at 260 nm. The extinction coefficient, ε_260_, was taken as 6600 mol^−1^·L cm^−1^ [50]. The stock solutions were stored at 4 °C and used after no more than 4 days. The metal complexes were dissolved in DMSO. The final concentration of DMSO in the buffered solution never exceeded 5%. The effects of the metal complexes on the DNA secondary structure were studied by keeping the concentration of CT-DNA at 5 × 10^−5^ mol∙L^−1^ [bp], while varying the concentration of the complex in the abovementioned buffer solution (r = [complex]/[DNA] = 0, 0.25, 0.5, 1.0). All CD spectra were recorded in the wavelength range 230 to 320 nm.

### 4.6. Melting Temperature Determination

All melting measurements were carried out in the solutions described for CD titrations experiments. DNA (45 µmol L^−1^) was then treated with our compounds at mol/bp ratio (r) = 0.1, and each sample was incubated for 24 h at room temperature. Samples were continuously heated with 1 °C min^−1^ rate of temperature increase, while the absorbance change at 260 nm was monitored. The investigated interval of temperature ranged from 50 to 90 °C. Upon reaching 90 °C, samples were cooled back to 50 °C in order to follow the renaturation process. Values for melting temperature (Tm) and for the melting interval (∆T) were determined according to the reported procedures [51].

Differential melting curves were obtained by numerical differentiation of experimental melting curves.

### 4.7. Statistical Analysis

The normality of data distributions was tested throughout the Kolmogorov–Smirnov test.

A two-way ANOVA was used to determine significant differences among treatments. Bonferroni post hoc test was used for pairwise comparisons. Statistical tests were performed using SPSS 25.0 software (IBM Corp. 2017).

## 5. Conclusions

In this work, the synthesis and the characterisation of three dimethylated thiosemicarbazone metal complexes ([Ni(4dm-(tcitr)_2_], Pt(4dm-(tcitr)_2_], and [Cu(4dm-(tcitr)_2_]) have been described. The metal complexes show a square planar geometry.

All investigated complexes were tested for in vitro antiproliferative potential against a human leukaemia cell line (U937). The GI_50_ values obtained after 24 h treatment for the metal complexes highlighted a mild and similar cytotoxic activity ([Pt(4dm-(tcitr)_2_] > [Cu(4dm-(tcitr)_2_] > [Ni(4dm-(tcitr)_2_]). On the contrary, the DNA damage observed by the Alkaline Comet Assay revealed that [Ni(4dm-(tcitr)_2_] is the most genotoxic compound. The Pt and Cu complexes caused DNA damage only at the highest tested concentrations.

In conclusion, these metals-based agents are suggested to be potentially good candidates as antiproliferative compounds. Future studies should deepen, on the one hand, their antiproliferative potential against a panel of normal and cancer cell lines deriving from different districts, and on the other, their action mechanism.

## Figures and Tables

**Figure 1 molecules-28-02778-f001:**
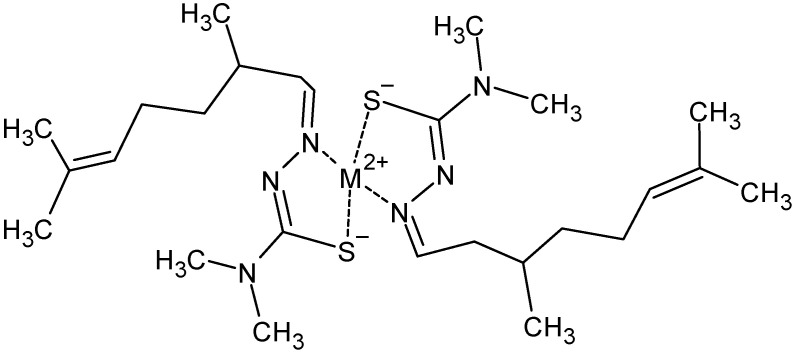
Schematic representation of the synthesised complexes.

**Figure 2 molecules-28-02778-f002:**
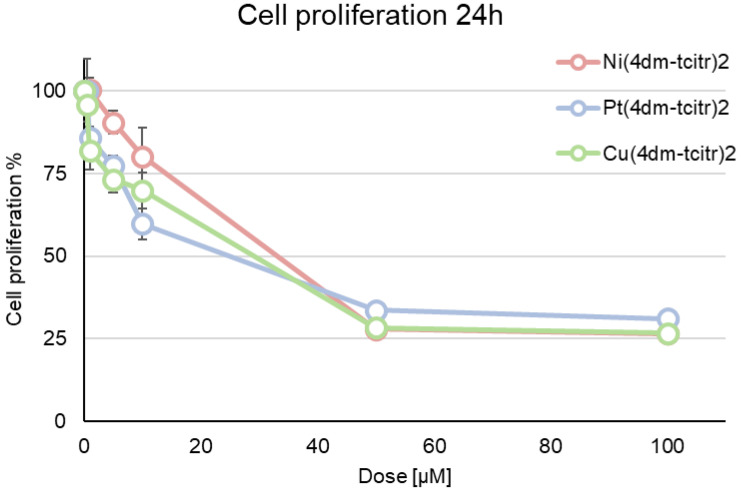
Dose–response curves obtained on U937 cells after 24 h treatment with [Ni(4dm-tcitr)_2_], [Pt(4dm-tcitr)_2_], and [Cu(4dm-tcitr)_2_]. Data are expressed as cell proliferation percentage compared with control cells.

**Figure 3 molecules-28-02778-f003:**
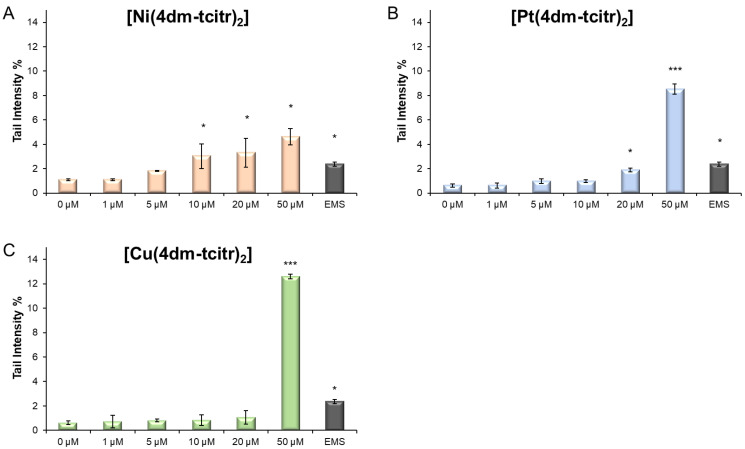
DNA damage detected by the Alkaline Comet Assay (pH > 13) in U937 cells treated with increasing doses of [Ni(4dm-tcitr)_2_] (**A**), [Pt(4dm-tcitr)_2_] (**B**), and [Cu(4dm-tcitr)_2_] (**C**) for 1 h. DNA damage is expressed as tail intensity percentage. * *p* < 0.05; *** *p* < 0.001.

**Figure 4 molecules-28-02778-f004:**
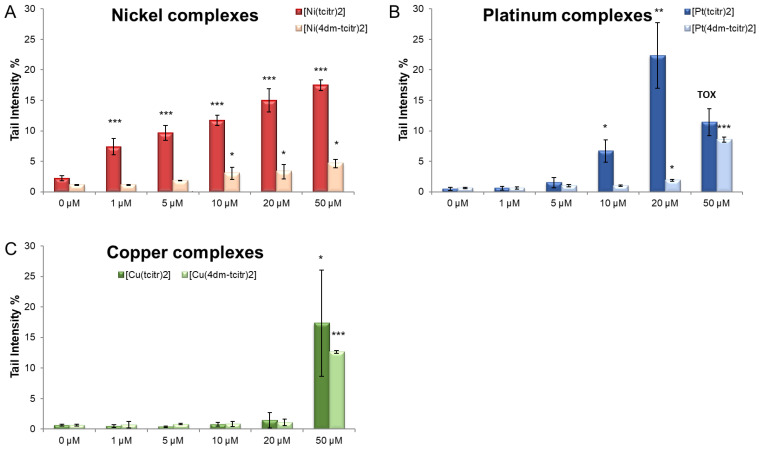
Comparison of genotoxic activity of parental ([Ni(tcitr)_2_] (**A**), [Pt(tcitr)_2_] (**B**), and [Cu(tcitr)_2_] (**C**)) and dimethylated compounds [Ni(4dm-tcitr)_2_] (**A**), [Pt(4dm-tcitr)_2_] (**B**), and [Cu(4dm-tcitr)_2_] (**C**). DNA damage was detected by the Alkaline Comet Assay (pH > 13) in U937 cells treated for 1 h with increasing concentrations of metal complexes. * *p* < 0.05; ** *p*< 0.01; *** *p* < 0.001.

**Figure 5 molecules-28-02778-f005:**
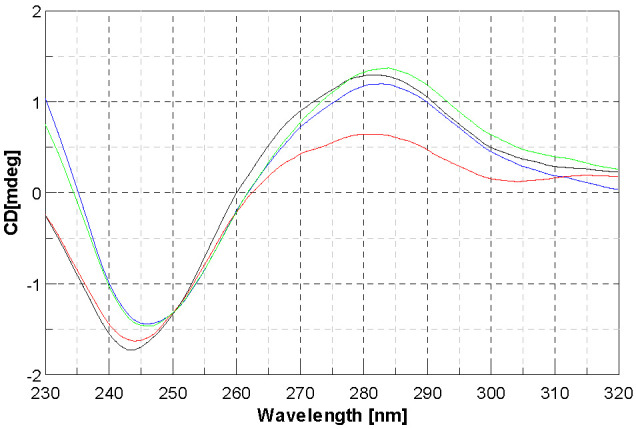
Circular dichroism spectra of CT-DNA with [Pt(4dm-(tcitr)_2_] at r = 0 (blue line), r = 0.25 (green line), r = 0.5 (back line), r = 1 (red line); (r = [complex]/[DNA]).

**Figure 6 molecules-28-02778-f006:**
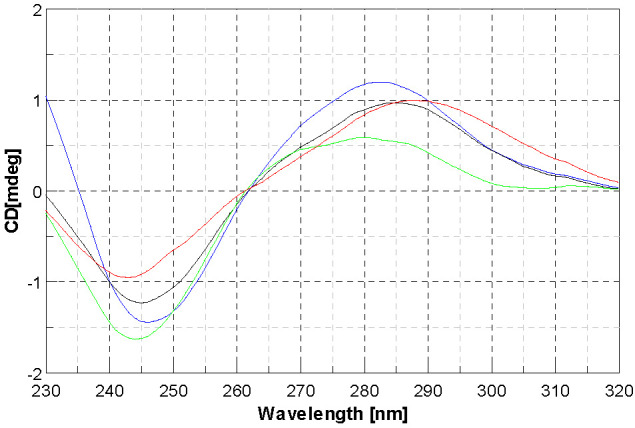
Circular dichroism spectra of CT-DNA with [Ni(4dm-(tcitr)_2_] at r = 0 (blue line), r = 0.25 (green line), r = 0.5 (back line), r = 1 (red line); (r = [complex]/[DNA]).

**Figure 7 molecules-28-02778-f007:**
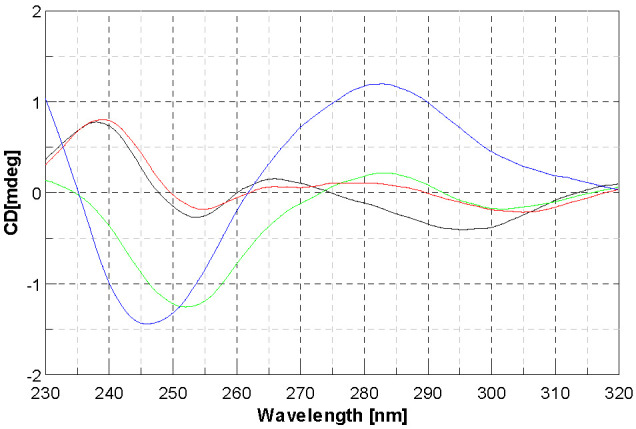
Circular dichroism spectra of CT-DNA with [Cu(4dm-(tcitr)_2_] at r = 0 (blue line), r = 0.25 (green line), r = 0.5 (back line), r = 1 (red line); (r = [complex]/[DNA]).

**Figure 8 molecules-28-02778-f008:**
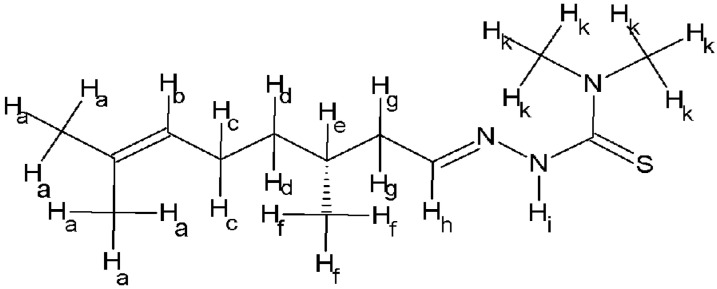
Schematic drawing of 4dm-Htcitr reporting the proton assignments for NMR interpretation.

**Table 1 molecules-28-02778-t001:** Comparison between the GI_50_ values (µM) obtained in cell line U937 after 24 h treatment with parental metal complexes ([Ni(tcitr)_2_], [Pt(tcitr)_2_], and [Cu(tcitr)_2_]) and dimethylated derivates ([Ni(4dm-tcitr)_2_], [Pt(4dm-tcitr)_2_], and [Cu(4dm-tcitr)_2_]). Data are expressed as GI_50_ values (µM), concentration of newly synthesised molecules inhibiting U937 cellular growth at 50% ± standard deviation (sd).

	GI_50_ Value (µM) ± sd
[Ni(4dm-tcitr)_2_]	33.5 ± 2.9
[Pt(4dm-tcitr)_2_]	25.3 ± 3.4
[Cu(4dm-tcitr)_2_]	30.2 ± 2.3
[Ni(tcitr)_2_]	10.0 ± 0.9
[Pt(tcitr)_2_]	7.0 ± 0.16
[Cu(tcitr)_2_]	33.0 ± 1.2

**Table 2 molecules-28-02778-t002:** Statistically significant differences among treatments obtained using SPSS 25.0 software (IBM-Italia, Segrate, Milano, Italy)). ANOVA, Bonferroni post hoc test, ** *p* ≤ 0.01; and *** *p* ≤ 0.001.

	DMSO	0.5 µM	1.0 µM	5.0 µM	10.0 µM	50.0 µM	100.0 µM
[Ni(4dm-tcitr)_2_]	-	-	-	-	***	***	***
[Pt(4dm-tcitr)_2_]	-	-	-	***	***	***	***
[Cu(4dm-tcitr)_2_]	-	-	**	***	***	***	***

## Data Availability

Not applicable.

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
