# Peer review of "Antiproliferative Activity and DNA Interaction Studies of a Series of N4,N4-Dimethylated Thiosemicarbazone Derivatives"

_molecules, 2023, doi:10.3390/molecules28062778_

Round 1

Reviewer 1 Report

General Comments

Thank you for sending the opinion manuscript entitled “Toxicological profile and DNA interaction studies of a series of dimethylated thiosemicarbazone derivatives” for possible publication in Molecules. The current work deals with new anticancer agents with novel modes of action may represent an innovative and successful strategy in the field of medicinal chemistry. I found, this work appears to be new and commendable for getting publication. However, it needs with mandatory minor corrections before getting accept for publication in this journal. Hence, I hereby inform that the manuscript recommended for publication to this journal.

Specific Comments

          The authors should include the abstract for brief understanding (background, methods, results and conclusions) of the readers before get into full article. I suggest not given detailed results part.

          Methods section: Why the authors chosen leukemia cell line (U937), in this study. Need to give proper reason and justification.

          Results: In figure 2, no data for control, the authors used only used molecules without control or standard drug for cell proliferation.

          Results. Line No. 148, Percentage of DNA, means not understand clearly, how it was calculated, need explanation. 

          Discussion. Why the authors included comparatives study in this discussion part. It should be include in the results part and proper discussion will be in the discussion part.

          Conclusion. It should separate section, need to explain elaborately with obtained results, significance and future prospective of the work.

          References: Not presented in the journal format and some the citrated references were old means before 2000, at least should cite recent relevant references. In addition, number of references also more than 78, possible reduce into below 50 numbers.

Author Response

Reviewer 1

General Comments

Thank you for sending the opinion manuscript entitled “Toxicological profile and DNA interaction studies of a series of dimethylated thiosemicarbazone derivatives” for possible publication in Molecules. The current work deals with new anticancer agents with novel modes of action may represent an innovative and successful strategy in the field of medicinal chemistry. I found, this work appears to be new and commendable for getting publication. However, it needs with mandatory minor corrections before getting accept for publication in this journal. Hence, I hereby inform that the manuscript recommended for publication to this journal.

Specific Comments

  • The authors should include the abstract for brief understanding (background, methods, results and conclusions) of the readers before get into full article. I suggest not given detailed results part.

We are very grateful for the time that the Reviewer spent for revising our manuscript. We checked and modified the abstract, consequently.

  • Methods section: Why the authors chosen leukemia cell line (U937), in this study. Need to give proper reason and justification.

We used human leukocytes U937 to define a toxicological profile of the tested compounds on human blood cells. In our previous works, we employed this cell model to investigate the antiproliferative activity and the action mechanism of newly synthesized molecules. Currently, these cells represent in our lab the first in vitro model for a preliminary screening of biological activity of new metal-based compounds. In addition, several literature studies report U937 cells as a reliable cell-based model to test drug toxicity and allergenic potential.

  • Results: In figure 2, no data for control, the authors used only used molecules without control or standard drug for cell proliferation.

In figure 2, dose 0 (cell proliferation = 100%) represents our negative control: cells were treated with our control vehicle, DMSO.

  • Line No. 148, Percentage of DNA, means not understand clearly, how it was calculated, need explanation.

In the Alkaline Comet Assay, the percentage of DNA is represented by the tail intensity percentage (TI%), a useful parameter to identify the genotoxic potential of the metal complexes. TI% was automatically calculated using a specific image analysis system (Comet Assay IV – Perceptive Instruments Ltd).

  • Why the authors included comparatives study in this discussion part. It should be include in the results part and proper discussion will be in the discussion part.

We followed the advice of the Reviewer: we moved the figure in the results section.

  • It should separate section, need to explain elaborately with obtained results, significance and future prospective of the work.

As highlighted by the Reviewer, we added a conclusion section to summarize our results and to anticipate future perspectives.

  • References: Not presented in the journal format and some the citrated references were old means before 2000, at least should cite recent relevant references. In addition, number of references also more than 78, possible reduce into below 50 numbers.

We took advice of the Reviewer, and we reduced the citated references. In addition, we checked and cited only papers after 2000, where possible.  

Reviewer 2 Report

The manuscript entitled “Toxicological profile and DNA interaction studies of a series of dimethylated thiosemicarbazone derivatives” may be of interest to the journal´s readers in some degree. The biological part is well done however in the chemical one some points need to be addressed and in my opinion the introduction is poor. Overall, I recommend a major revision of the manuscript.

Major Comments:

1.   Introduction. The authors focus the first paragraph of this section on the antitumor properties of the TSCs and it is why I can not understand what the second one goes into since in the manuscript neither other pharmacological properties are evaluated nor heterocyclic TSCs are used.

2.   Line 88-91. This sentence should be rewritten to describe only the new syntheses performed here“

3.   Figure 1. Are the authors sure of which atoms behave as donors? They have drawn two four-membered chelate rings which is not in agreement with analogous complexes obtained with similar ligands.

4.   Table 1 and Figure 2. Are two different ways of representing the same data, one of them should be removed.

5.   Table 3. Include all the data represented in Table 2. In my opinion this table should be removed, and explain the results of the experiment in the text including the corresponding reference.

6.   Line 379. The H1NMR data for [Cu(4dm-(tcitr)2] is mising.

Author Response

Reviewer 2

The manuscript entitled “Toxicological profile and DNA interaction studies of a series of dimethylated thiosemicarbazone derivatives” may be of interest to the journal´s readers in some degree. The biological part is well done however in the chemical one some points need to be addressed and in my opinion the introduction is poor. Overall, I recommend a major revision of the manuscript.

Major Comments:

  • The authors focus the first paragraph of this section on the antitumor properties of the TSCs and it is why I can not understand what the second one goes into since in the manuscript neither other pharmacological properties are evaluated nor heterocyclic TSCs are used.

Our work is aimed at the drug-design of molecules with potential antiproliferative activity, with a focus on TSCs. In the introduction we have tried to highlight this aspect which involves both the chemical and biological sectors.

  • Line 88-91. This sentence should be rewritten to describe only the new syntheses performed here“

We thank the Reviewer for the observation. We modified the sentence accordingly. 

  • Figure 1. Are the authors sure of which atoms behave as donors? They have drawn two four-membered chelate rings which is not in agreement with analogous complexes obtained with similar ligands.

We thank the Reviewer. The figure 1 was wrong! We added in the manuscript a new figure 1.

  • Table 1 and Figure 2. Are two different ways of representing the same data, one of them should be removed.

In the table 1, we summarized all the GI50 value obtained after 24 h of treatment for the newly synthesized metal complexes and for the parental compounds, previously reported in the table 3. In the figure 2 we showed the dose-response curve for each novel compound.

Table 3. Include all the data represented in Table 2. In my opinion this table should be removed, and explain the results of the experiment in the text including the corresponding reference.

As recommended by the Reviewer, all the GI50 values of parental and derivate compounds were summarized in the table 1.

  • Line 379. The H1NMR data for [Cu(4dm-(tcitr)2] is mising.

[Cu(4dm-(tcitr)2] is a Cu(II) complex and therefore paramagnetic. That’s the reason why we did not perform its 1H NMR characterization. The oxidation state of the metal ion should be clearer now with the correct figure 1.

Reviewer 3 Report

Paper entitled Toxicological profile and DNA interaction studies of a series of
dimethylated thiosemicarbazone derivatives is a certain follow up on a work already performed by the same group, and thus gives some novel insight into activity of some novel compounds. However, although authors claim series of derivates (I am not sure that the noun used in the title is correct!), present only few in a paper.

I am not sure about synthesis protocol, but please consider some better explanation other than "The Cu(II), Ni(II) and Pt(II) analogues were isolated".

In addition, although toxicology profile is anticipated, in the paper one can not find toxicology screening, except genotoxic test on leukemia cell line! Please consider what is the aim of the present paper (not your study in total) and revise the title accordingly.

Please state where is the difference between this paper and paper reference 32.

I strongly suggest, in order to address the anticancer potential of derivates to perform same experiments on other cancer cell line as well on "normal" cell line (fibroblasts, erithrocites, leucocytes...). If these compounds are not, at least partially selective, than they are highly toxic, based on your results.

Minor:

Please reduce the number of references, especially in introduction section. For example: "particularly with nickel [30-33]; copper [34-39]; platinum [40-42]; zinc [43,44]; gold [45,46])". Here I am sure you can reduce them. But elswhere as well.

Author Response

Reviewer 3

Paper entitled Toxicological profile and DNA interaction studies of a series of dimethylated thiosemicarbazone derivatives is a certain follow up on a work already performed by the same group, and thus gives some novel insight into activity of some novel compounds. However, although authors claim series of derivates (I am not sure that the noun used in the title is correct!), present only few in a paper.

To avoid misunderstandings, we have added an indication of the position of the substituents in the title.

  • I am not sure about synthesis protocol, but please consider some better explanation other than "The Cu(II), Ni(II) and Pt(II) analogues were isolated".

The sentence, that appeared concise in the abstract due to character number restrictions, has been modified. In any case, a detailed synthesis protocol is described in the 4.1. paragraph.

  • In addition, although toxicology profile is anticipated, in the paper one can not find toxicology screening, except genotoxic test on leukemia cell line! Please consider what is the aim of the present paper (not your study in total) and revise the title accordingly.

We decided according to this comment to change the title of the paper in “Antiproliferative activity and DNA interaction studies of a series of N4,N4-dimethylated thiosemicarbazone derivatives”

  • Please state where is the difference between this paper and paper reference 32.

In the paper reference 32 (in the revised manuscript the corresponding reference is 20), we described for the first time the antiproliferative, genotoxic and mutagenic activities of nonmethylated platinum and copper compounds ([Pt(tcitr)2] and [Cu(tcitr)2]). In addition, we deepened the mechanism of action of [Ni(tcitr)2], [Pt(tcitr)2] and [Cu(tcitr)2] to investigate whether their biological activity can be ascribed to a direct interaction with DNA. Here, we presented a new series of dimethylated metal-based compounds ([Ni(4dm-(tcitr)2], Pt(4dm-(tcitr)2] and [Cu(4dm-(tcitr)2]) and their chemical synthesis and characterization. We reported their cyto- and genotoxic potential and their ability to interact with DNA.

  • I strongly suggest, in order to address the anticancer potential of derivates to perform same experiments on other cancer cell line as well on "normal" cell line (fibroblasts, erithrocites, leucocytes...). If these compounds are not, at least partially selective, than they are highly toxic, based on your results.

We agree with the Reviewer about the importance to perform additional tests on a panel of cancer and normal cells deriving from different districts. We are aware that the requirement for more tests is necessary before considering these new molecules as potential drugs according to current regulations. Our approach to test their efficacy and toxicity is only a first phase of their evaluation. In a second phase of evaluation of the most interesting compounds, we plan to investigate the toxicological activity on different normal tissue in vitro and in vivo.

Minor:

  • Please reduce the number of references, especially in introduction section. For example: "particularly with nickel [30-33]; copper [34-39]; platinum [40-42]; zinc [43,44]; gold [45,46])". Here I am sure you can reduce them. But elswhere as well.

We thank the Reviewer for having underlined this issue and we reduced the citated references, consequently.

Round 2

Reviewer 2 Report

Overall, the authors addressed most of my concerns satisfactorily.

Reviewer 3 Report

.